# Prevalence of diabetes in pregnancy and microvascular complications in native Indonesian women: The Jogjakarta diabetic retinopathy initiatives in pregnancy (Jog-DRIP)

**Felicia Widyaputri**[1,2,3], **Lyndell L. Lim**[2,3,4], **Tiara Putri Utami**[1], **Annisa Pelita Harti**[5], **Angela Nurini Agni**[1], **Detty Siti Nurdiati**[5], **Tri Wahyu Widayanti**[1], **Supanji**[1], **Firman Setya Wardhana**[1], **Mohammad Eko Prayogo**[1], **Muhammad Bayu Sasongko**[1]*

1 Department of Ophthalmology, Faculty of Medicine, Public Health and Nursing, Universitas Gadjah Mada, Yogyakarta, Indonesia, 2 Centre for Eye Research Australia, Royal Victorian Eye and Ear Hospital, Melbourne, Australia, 3 Ophthalmology, Department of Surgery, University of Melbourne, Melbourne, Australia, 4 Royal Victorian Eye and Ear Hospital, Melbourne, Australia, 5 Department of Obstetrics and Gynecology, Faculty of Medicine, Public Health and Nursing, Universitas Gadjah Mada, Yogyakarta, Indonesia

* mb.sasongko@ugm.ac.id

## Abstract

### Objectives

To report the prevalence of total diabetes in pregnancy (TDP) and diabetes-related microvascular complications among Indonesian pregnant women.

### Methods

We conducted a community-based cross-sectional study with multi-stage, cluster random sampling to select the participating community health centers (CHC) in Jogjakarta, Indonesia between July 2018-November 2019. All pregnant women in any trimester of pregnancy within the designated CHC catchment area were recruited. Capillary fasting blood glucose (FBG) and blood glucose (BG) at 1-hour (1-h), and 2-hour (2-h) post oral glucose tolerance test (OGTT) were measured. TDP was defined as the presence of pre-existing diabetes or diabetes in pregnancy (FBG ≥7.0 mmol/L, or 2-h OGTT ≥11.1 mmol/L, or random BG ≥11.1 mmol/L with diabetes symptoms). Disc and macula-centered retinal photographs were captured to assess diabetic retinopathy (DR). Blood pressure, HbA1c and serum creatinine levels were also measured.

### Results

A total of 631/664 (95%) eligible pregnant women were included. The median age was 29 (IQR 26–34) years. The prevalence of TDP was 1.1% (95%CI 0.5, 2.3). It was more common in women with chronic hypertension (p = 0.028) and a family history of diabetes (p = 0.015). Among the TDP group, 71% had a high HbA1c, but no DR nor nephropathy were observed.

**Data Availability Statement:** All relevant data are within the manuscript and its Supporting Information files.

**Funding:** This study was supported by the Australia-Indonesia Institute in the Australian Government's Department of Foreign Affairs and Trade in the form of funding awarded to MBS (AII2018/19073), and the Indonesian Endowment Fund for Education (LPDP) Ministry of Finance in the form of a grant awarded to FW (20161012049462). The funders had no role in study design, data collection and analysis, decision to publish, or preparation of the manuscript.

**Competing interests:** The authors have declared that no competing interests exist.

## Conclusions

Although a very low prevalence of TDP and no diabetes-related microvascular complications were documented in this population, there is still a need for a screening program for diabetes in pregnancy. Once diabetes has been identified, appropriate management can then be provided to prevent adverse outcomes.

## Introduction

Diabetes mellitus (DM) is one of the most important non-communicable diseases worldwide causing 4.8 million deaths, significant morbidities, and permanent disabilities every year [1]. Whilst there is an extensive body of literature regarding the prevalence, incidence, complications and state-of-the-art treatments for this condition, diabetes during pregnancy is particularly challenging due to the complexity of its management [2–5].

A new categorization of diabetes during pregnancy, now known as hyperglycemia in pregnancy, has been proposed following the results of the Hyperglycemia and Adverse Pregnancy Outcome (HAPO) study [6–8]. Hyperglycemia in pregnancy is sub-categorized into two types: total diabetes in pregnancy (TDP) and gestational diabetes mellitus (GDM) [4, 9]. TDP includes diabetes that is first diagnosed in pregnancy with glucose levels fulfilling the criteria of diabetes in non-pregnant adults (previously undiagnosed diabetes) and pre-existing diabetes mellitus (PDM) in pregnancy (either type 1 or type 2 diabetes diagnosed prior to pregnancy) [9], whereas GDM is defined as a milder degree of glucose intolerance that is first recognized during pregnancy [7].

Total diabetes in pregnancy is clinically more important because it is associated with a higher risk of severe pregnancy complications that can affect both the mother and the offspring than GDM [7]. Importantly, among pregnant women with TDP, there is thought to be significant risk diabetes-related complications (i.e., diabetic retinopathy and other microvascular complications) worsening during pregnancy that will persist for a lifetime, beyond the pregnancy itself [8]. Therefore, careful management, including screening for diabetes complications, is crucial to prevent adverse outcomes for the mother and baby.

This study aimed to report the prevalence of TDP, diabetic retinopathy (DR) and other microvascular complications in an Indonesian population. While TDP is known to be prevalent amongst countries in Western Pacific Region, information regarding the prevalence of TDP in Indonesian population, which is one of 10 countries with the highest number of people with diabetes and undiagnosed diabetes globally, is currently lacking, highlighting the importance of this study [4, 9].

## Research design and methods

### Study design, sampling methods and study population

This was a community-based cross-sectional study. We recruited pregnant women in any trimester of pregnancy who resided in Jogjakarta between July 2018 and November 2019. This study was conducted following the Declaration of Helsinki. The ethical clearance, information sheet, and consent form were approved by the Medical and Health Research Ethics Committee (MHREC), Faculty of Medicine, Universitas Gadjah Mada–Dr. Sardjito General Hospital. Written consents to participate in this study and to report all relevant data in the subsequent

publications were obtained from all participants, and for those who were unable to see, read, or write, verbal consents were obtained.

Jogjakarta is one of the most densely populated provinces in Indonesia. There are approximately 3.8 million residents living in the province, with a population density of 1,199 people/km². This area had approximately 49,000 pregnancies and 43,000 live births annually [10]. The recruitment strategy and sampling approach used in this study were similar to our previous study [11, 12]. In brief, Jogjakarta has four regencies and a municipality with 121 CHCs spread throughout the area. Multi-stage, clustered random sampling was used to determine 21 community health centers (CHCs) as primary recruitment sites, to provide good representation of the province (Fig 1). Due to logistic constraints and the requirement to attend Dr. Sardjito

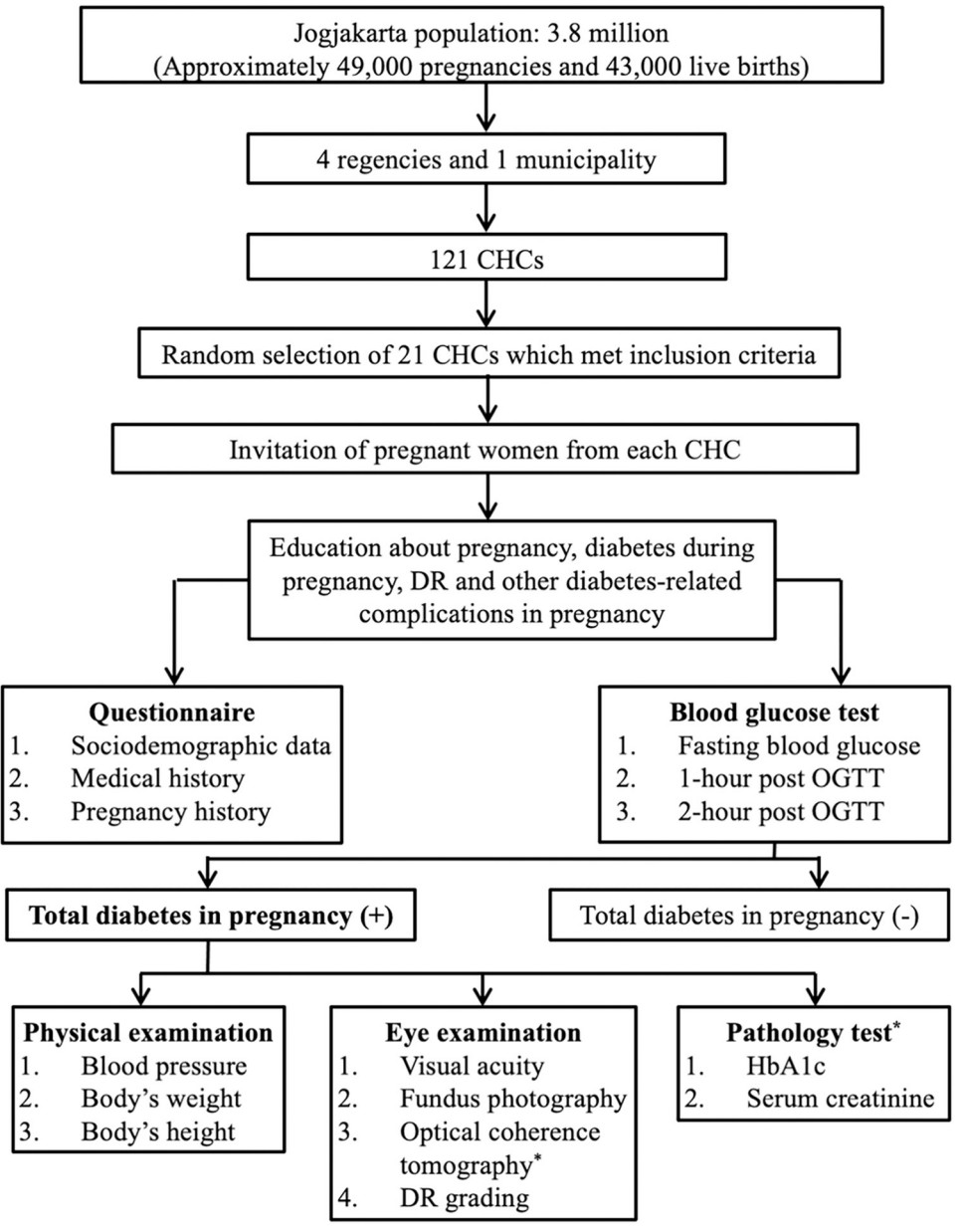

**Fig 1. Jogjakarta Diabetic Retinopathy Initiatives in Pregnancy (Jog-DRIP) study design.** CHC, community health center; DR, diabetic retinopathy; HbA1c, glycated hemoglobin; OGTT, oral glucose tolerance test. *Performed at the Dr. Sardjito General Hospital.

General Hospital for further pathology tests, we only included CHCs that were located within 60 kms from the hospital.

Sample size was estimated using the formula for prevalence studies with multi-stage cluster sampling (approximate number of pregnancies in Jogjakarta: 49,000 [10], using the precision of estimates to identify prevalence of hyperglycemia in pregnancy at 27% [4]). A minimum sample size of 630 pregnant women (around 30 from each cluster) was sufficient to demonstrate the prevalence with sufficient power and precision (95% confidence interval [95% CI] 21%, 32%).

This study was fully supported by the province's local health authorities (LHA) and LHAs in each regency/ municipality. Initially, all family physicians (FPs) and midwives from CHCs in each region were invited by their LHA to attend a workshop that provided information about hyperglycemia in pregnancy, DR, screening, and examination protocols. After the workshop, FPs and midwives invited all pregnant women listed in their registry to attend the diabetes and DR screening session conducted by our research team. There were 664 pregnant women screened from 21 CHCs. We excluded 33 pregnant women due to failure to fast for a minimum of 8 hours, leaving 631 eligible pregnant women for hyperglycemia testing (95%).

## Assessment of hyperglycemia

All participants underwent an oral glucose tolerance test (OGTT) using 75-g of anhydrous glucose dissolved in 250 ml water after overnight fasting (8–14 hours of no caloric intake). Capillary fasting blood glucose (FBG), 1-hour (1-h), and 2-hour (2-h) post-OGTT blood glucose levels were measured using a blood glucose meter (Roche Accu-Check Performa II, Roche Diabetes Care, Indiana, United States) by trained examiners. Capillary blood glucose (CBG) was used due to the limited laboratory access in the CHCs. For participants who were diagnosed with diabetes before their pregnancy, diabetes diagnosis was confirmed by their family physician following the American Diabetes Association criteria [2], thus; OGTT was not performed. This study used "pre-existing diabetes (PDM)" to describe diabetes that was diagnosed prior to pregnancy, "Diabetes in Pregnancy (DIP)" to describe diabetes first diagnosed in pregnancy (having one of the following: FBG ≥7.0 mmol/L, or 2-H OGTT ≥11.1 mmol/L, or random BG ≥11.1 mmol/L in the presence of diabetes symptoms), and "Total Diabetes in Pregnancy (TDP)" to describe both PDM and DIP (as shown in S1 Fig) [7]. A 20 mg/dl (1.11 mmol/L) addition to our 1-h and 2-h post-OGTT values was applied for the conversion from CBG used in our study to venous plasma glucose (VPG) as recommended in the guidelines [13, 14]. During the oral glucose load, 13 women vomited and refused to continue with the OGTT. For these women, the diabetes status was determined only based on their FBG level.

## Eye examinations and diabetic retinopathy assessment

Only participants who had TDP underwent further eye examinations. Visual acuity was assessed using a Snellen Chart or E-chart at a distance of 6 meters by senior ophthalmology residents. Two-field (disc and macula-centered) fundus photographs and OCT scans were captured to assess the presence of DR and diabetic macular oedema (DMO) using a MiiS Horus + Scope DEC 200 Eye Fundus Camera (Medimaging Integrated Solution Inc., Taiwan) and a Stratus OCT™ 3000 (Carl Zeiss Meditec Inc., USA), respectively, without pupil dilatation.

DR severity was graded from fundus photographs by a trained grader masked to women's clinical details. DR was categorized into five severities based upon the Modified Airlie House Classification system as follows: 1) No DR including Early Treatment Diabetic Retinopathy Study (ETDRS) levels 10 and 12; 2) mild non-proliferative DR (NPDR) including ETDRS levels 14 to 20; 3) moderate NPDR including ETDRS levels 31 and 41; 4) severe NPDR including

ETDRS levels 51 to 53; and 5) proliferative DR (PDR) including ETDRS levels 61 to 80 [15]. The diagnosis of DMO was based upon quantitative data on the OCT scan. Central sub-field thickness (CSFT) from the ETDRS grid centered on the macula that were generated by the built-in software within OCT devices was collected. The presence of DMO was defined as having CSFT above the normative value (>239 μm) [16].

## Other clinical examinations

A structured questionnaire was developed and validated prior to the commencement of the study. We used this pre-developed questionnaire to obtain all data relevant to demographic characteristics (age, level of education, average income per month, and health insurance coverage), pregnancy details (gestational age and history of previous pregnancy), and general medical histories (weight before pregnancy, gravidity, history of diabetes, hypertension, dyslipidemia, smoking, and family history of diabetes).

All clinical examinations, including blood pressure (BP) and body mass index (BMI), were performed by a study field coordinator with a general medicine qualification and license to practice as a family physician. BP measurement was done using an automated BP monitor (Omron Arm Blood Pressure Monitor JPN-500, Omron Healthcare, Inc., Kyoto, Japan) and repeated three times for each participant. Hypertension in pregnancy was defined as having either systolic BP ≥140 mmHg or diastolic BP ≥90 mmHg or any history of receiving treatment with BP-lowering medications [17]. Pre-pregnancy BMI was calculated from the self-reported weight before pregnancy and their measured height and categorized as underweight (<18.5 kg/m$^2$), normal (18.5–24.9 kg/m$^2$), overweight (25–29.9 kg/m$^2$), or obese (≥30 kg/m$^2$) [18].

Participants with confirmed TDP were invited to come to Dr. Sardjito General Hospital for additional blood tests to assess other microvascular complications. Venous blood samples were collected in EDTA vacutainer tubes and tested at the Sardjito Hospital Laboratory for evaluation of HbA1c and serum creatinine levels. A high HbA1c was defined as having HbA1c level ≥6.5% or ≥48 mmol/mol, which is associated with a higher risk of diabetes-related complications, including DR, while nephropathy was defined as a serum creatinine level >77 μmol/L or >0.87 mg/dl [19–21].

## Statistical analyses

Statistical analyses were carried out using Stata IC 16.1 for Mac (College Station, TX, USA). Descriptive statistics, including frequencies and percentages, were obtained. Data distribution normality was tested using a Shapiro Wilk test. Prevalence of TDM, PDM, and DIP were estimated by dividing the number of cases with the total number of included women, with 95% confidence intervals (95%CIs) for these rates calculated using Agresti-Coull method or Wilson score interval, as appropriate. Demographic characteristics were compared between TDP group and no TDP group (including pregnancies without diabetes and with GDM) using the Wilcoxon rank-sum test or Fisher's exact test. Characteristics comparison between pregnancies with TDP, GDM, and without diabetes can be seen in the S1 Table. Among participants with TDM, the rate of diabetes-related complications (e.g., DR, hypertension in pregnancy, nephropathy) and a high HbA1c level was determined.

## Results

Of the 664 pregnant women who attended the screening examination, 631 (95%) were included in the final analysis. The median maternal age was 29 years (Interquartile range [IQR] 26, 34). Overall, 5.0%, 9.9%, and 13% of our cohort had a history of hypertension and

dyslipidemia before pregnancy, and had at least one immediate family member with diabetes, respectively. The majority of women resided in the urban area (76%) and had health insurance (74%). Among 340 women with previous pregnancy data, 25 (7.3%) women had a baby with a birth weight of more than 4.0 kgs (macrosomia).

The overall prevalence of TDP in our study population was 1.1% (95% CI 0.5, 2.3), consisting of DIP 0.6% (95% CI 0.2, 1.7) and PDM 0.5% (95% CI 0.1, 1.5). There were three cases of PDM: all had type 2 diabetes, with a diabetes duration of one month, six months, and 36 months. The comparison of women's characteristics in TDP and no TDP groups was presented in Table 1. Pregnant women with TDP were more likely to have a history of hypertension ($p$ = 0.043) and a family history of diabetes ($p$ = 0.017) compared to those with normal glucose tolerance. No other significant differences were found between the two groups. Among women with TDP, 71% (95% CI 35.9, 91.8) had high HbA1c, and 20% (95% CI 3.62, 62.45) had hypertension in pregnancy. However, none of the participants were found to have either DR or nephropathy. The median CSFT of participants with TDP was 236.5 μm (IQR 232.5–243).

## Discussion

The current study documented that the overall prevalence of TDP among pregnant women in an Indonesian population was 1.1%; around 57% was due to DIP (equal to 0.6% of the study population). The presence of TDP was more common in pregnant women with chronic hypertension or those with family history of diabetes; however, there were no DR nor nephropathy observed in this population.

Our study was the first to report the prevalence of TDP among pregnant women in Indonesian population since the introduction of the WHO 2013 criteria. Moreover, there were no studies that have reported data on the proportion of DIP and PDM. The prevalence of TDP documented in our population was comparable to the prevalence in China and Hongkong (1.5%), Western Pacific (WP) Region (1.7%) and Europe (1.7%) [9]. A higher prevalence of TDP was reported from Middle East and North Africa Region (4.0%) and also Malaysia (4.65%) [9]. This is interesting because a prior study has reported a lower rate of diabetes and undiagnosed diabetes in the general Malaysian population than in Indonesia [4]. One possible explanation may be that TDP prevalence in the Malaysian report was estimated using data from two hospital-based studies that could have captured more pregnant women with TDP who were mostly referred to the hospital for antenatal care. This contrasts with the design of our study, which was community-based and therefore more likely to reflect the true prevalence within the general population.

We did not document any evidence of DR in this study cohort. A study by Sugiyama and associates also reported that the prevalence of DR among pregnant women in Japan was very low, where only 1.2% of pregnant women with overt diabetes (which had a similar definition with the current DIP) had DR [22]. In contrast, Rasmussen and colleagues, who studied Danish pregnant women with type 2 diabetes, showed that DR was significantly higher, at 14% of 110 pregnant women [23]. This discrepancy might be due to the shorter duration of diabetes in our study (a median of 6 months) compared to Rasmussen's study (a mean of 3.3 years for the no progression group and 6.7 years for the progression group). Another reason might be due to the small number of TDP cases in our cohort that decreased our ability to detect DR. Further study involving a bigger number of TDP cases are needed to confirm our findings. Apart from DR, we also documented no case of nephropathy or kidney dysfunction. This is in line with previous reports that showed low rates of nephropathy in this population and that pregnancy was less likely to induce diabetic nephropathy [24, 25]. Comparing our findings

**Table 1. Demographic characteristics of study participants with and without total diabetes in pregnancy.**

| Characteristics | Overall | No TDP | TDP | p-value[a] |
|---|---|---|---|---|
| **N** | 631 | 624 | 7 | |
| **Age group** [years], n (%) | | | | 0.220 |
| <= 25 | 148 (24.50) | 147 (24.62) | 1 (14.29) | |
| 26–35 | 352 (58.28) | 349 (58.46) | 3 (42.86) | |
| >= 35 | 104 (17.22) | 101 (16.92) | 3 (42.86) | |
| **Age** [years], median (IQR) | 29 (26, 34) | 29 (26, 34) | 34 (26, 39) | 0.252 |
| **Pregnancy stage at screening**, n (%) | | | | 0.320 |
| First trimester | 73 (11.85) | 71 (11.66) | 2 (28.57) | |
| Second trimester | 341 (55.36) | 338 (55.50) | 3 (42.86) | |
| Third trimester | 202 (32.79) | 200 (32.84) | 2 (28.57) | |
| **Gestational age at screening** [weeks], median (IQR) | 25 (20, 29) | 25 (20, 29) | 23 (9, 32) | 0.676 |
| **Gravidity**, n (%) | | | | 0.829 |
| Prime | 210 (34.77) | 207 (34.67) | 3 (42.86) | |
| Second and third | 352 (58.28) | 348 (58.29) | 4 (57.14) | |
| Fourth and above | 42 (6.95) | 42 (7.04) | 0 | |
| **Level of education,** n (%) | | | | 0.767 |
| Never went to school | 9 (1.49) | 9 (1.51) | 0 | |
| Primary school | 27 (4.48) | 27 (4.53) | 0 | |
| Secondary school | 448 (74.30) | 443 (74.33) | 5 (71.43) | |
| University degree | 119 (19.73) | 117 (19.63) | 2 (28.57) | |
| **Household income/month** [IDR], n (%) | | | | 1.000 |
| <1,000,000 | 86 (23.82) | 84 (23.73) | 2 (28.57) | |
| 1,000,000–2,499,999 | 201 (55.68) | 197 (55.65) | 4 (57.14) | |
| 2,500,000–4,999,999 | 61 (16.90) | 60 (16.95) | 1 (14.29) | |
| >= 5,000,000 | 13 (3.60) | 13 (3.67) | 0 | |
| **Residence**, n (%) | | | | 0.465 |
| Urban area | 458 (75.83) | 452 (75.71) | 6 (85.71) | |
| Rural area | 146 (24.17) | 145 (24.29) | 1 (14.29) | |
| **BMI pre-pregnancy**, n (%) | | | | 0.212 |
| Underweight | 51 (9.32) | 51 (9.41) | 0 | |
| Normal | 312 (57.04) | 310 (57.20) | 2 (40.00) | |
| Overweight | 129 (23.58) | 128 (23.62) | 1 (20.00) | |
| Obesity | 55 (10.05) | 53 (9.78) | 2 (40.00) | |
| **BMI pre-pregnancy**, median (IQR) | 23.44 (20.81, 26.35) | 23.43 (20.81, 26.31) | 26.27 (22.21, 30.49) | 0.308 |
| **Systolic BP** [mmHg], median (IQR) | 110 (100, 120) | 110 (100, 120) | 115 (111, 129) | 0.136 |
| **Diastolic BP** [mmHg], median (IQR) | 72 (68, 80) | 72 (68, 80) | 70 (64, 80) | 0.889 |
| **Past smoker** [yes], n (%) | 20 (3.37) | 20 (3.41) | 0 | 0.786 |
| **Health insurance** [yes], n (%) | 271 (73.64) | 267 (73.96) | 4 (57.14) | 0.271 |
| **History of medical conditions**: | | | | |
| Hypertension [yes], n (%) | 30 (4.97) | 28 (4.69) | 2 (28.57) | 0.043 |
| Dyslipidaemia [yes], n (%) | 60 (9.93) | 60 (10.05) | 0 | 0.479 |
| Macrosomia baby in previous pregnancy [yes], n (%) | 25 (7.35) | 25 (7.44) | 0 | 0.736 |
| **Family history of diabetes** [yes], n (%) | 72 (12.79) | 69 (12.37) | 3 (60.00) | 0.017 |

BMI, body mass index; BP, blood pressure; IQR, interquartile range; TDP, total diabetes in pregnancy.

[a] p-value was estimated using Wilcoxon rank-sum test or Fisher's exact test as appropriate.

with the Indonesian non-pregnant population, the DiabCare Indonesia 2008 study found that within their participant group with diabetes duration of less than one year, 1 out of 15 participants had NPDR and no diabetic nephropathy was observed [26]. Although the DiabCare's cohort was older than ours, among those whose diabetes was less than one year, the rate of DR and diabetic nephropathy was similar compared to our study.

Our findings showed that pregnant women with TDP were more likely to have chronic hypertension and family history of diabetes. These results were consistent with findings from previous studies [27, 28]. A family history of diabetes was associated with an increased risk of diabetes in the non-pregnant population [29]. Similarly, an increased risk of GDM was also associated with family history of diabetes [27, 28]. Regarding chronic hypertension, an Iranian study found that a significantly higher proportion of pregnant women with chronic hypertension had GDM [28].

The strengths of this study were its community-based design, which resulted in more representative sampling of our population of native-Indonesian pregnant women, the involvement of family physicians and midwives from CHCs who played a role as the primary health career of our pregnant population, and the implementation of standardized examinations to assess DR and nephropathy. However, there were also several limitations. Firstly, the nature of our cross-sectional design limited the interpretation of our results. Secondly, we used CPG instead of VPG (the gold standard) for the screening examination. However, CPG has been proven to be acceptable for diabetes screening in pregnancy areas with limited resources [13] and was used in a similar study in Jakarta, Indonesia [30]. Thirdly, due to the small number of TDP cases detected in our study population, this study could have been underpowered to measure a precise rate of DR and nephropathy in pregnant women with TDP. Sample size estimation for future studies should consider small events of TDP and diabetes complications among TDP. Finally, due to the impact of COVID-19 pandemic throughout 2020, postpartum follow-up examinations could not be completed; thus, the outcome of the diabetes status of women with DIP and DR and nephropathy status in those with TDP after delivery could not be determined. Future studies with long period of follow-up covering post-partum period and the subsequent pregnancies would capture better picture related to the course of TDP and development of diabetes complications among TDP.

In summary, we documented that the prevalence of TDP in Indonesian pregnant women was very low, and similar to that estimated in Western Pacific Region. We further found that approximately 6 every 1000 Indonesian pregnant women had probable undiagnosed diabetes. The presence of TDP in Indonesian women were more common among those with chronic hypertension and family history of diabetes. Although minimal rate of diabetes-related microvascular complications was observed during pregnancy in this study, a national screening program for diabetes during pregnancy is needed to detect those with undiagnosed diabetes and at most risk of developing diabetes complications so that proper management can be introduced to minimize adverse outcomes to mothers and their future babies.

## Supporting information

**S1 Fig. Terminology and classification of hyperglycemia in pregnancy (adapted from Guariguata et al., 2013 [9]).**
(TIFF)

**S1 Table. Comparison of demographic characteristics between normal, GDM and TDP groups.**
(PDF)

**S1 Appendix. Patient questionnaire (English version).**
(PDF)

**S2 Appendix. Patient questionnaire (Indonesian version).**
(PDF)

## Acknowledgments

The authors thank Local Health Authority (LHA) of Jogjakarta Province and all involved CHCs for technical assistance.

## Author Contributions

**Conceptualization:** Felicia Widyaputri, Lyndell L. Lim, Muhammad Bayu Sasongko.

**Data curation:** Tiara Putri Utami, Annisa Pelita Harti.

**Formal analysis:** Felicia Widyaputri.

**Funding acquisition:** Felicia Widyaputri, Muhammad Bayu Sasongko.

**Investigation:** Tiara Putri Utami, Annisa Pelita Harti.

**Methodology:** Felicia Widyaputri, Annisa Pelita Harti, Detty Siti Nurdiati, Muhammad Bayu Sasongko.

**Project administration:** Tiara Putri Utami.

**Supervision:** Muhammad Bayu Sasongko.

**Visualization:** Felicia Widyaputri.

**Writing – original draft:** Felicia Widyaputri.

**Writing – review & editing:** Lyndell L. Lim, Tiara Putri Utami, Angela Nurini Agni, Detty Siti Nurdiati, Tri Wahyu Widayanti,  Supanji, Firman Setya Wardhana, Mohammad Eko Prayogo, Muhammad Bayu Sasongko.

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
