## [Decision Letter · Decision Letter 0]

16 Aug 2021

PONE-D-21-21657

Prevalence of total diabetes in pregnancy, diabetic retinopathy, and other microvascular complications in pregnancy in native Indonesian women: the Jogjakarta diabetic retinopathy initiatives in pregnancy (Jog-DRIP)

PLOS ONE

Dear Dr. Sasongko,

Thank you for submitting your manuscript to PLOS ONE. After careful consideration, we feel that it has merit but does not fully meet PLOS ONE’s publication criteria as it currently stands. Therefore, we invite you to submit a revised version of the manuscript that addresses the points raised during the review process.

I have received the reports from our advisors on your manuscript which you submitted to PLOS ONE.

Based on the comments received, I feel that your manuscript could be reconsidered for publication should you be prepared to incorporate major revisions.

When preparing your revised manuscript, you are asked to carefully consider the reviewer comments below and submit a list of responses to the comments.

Editor Comments: The paper should be checked by a professional speaker of English before complete acceptance.

We look forward to receiving your revised manuscript.

Kind regards,

Muhammad Sajid Hamid Akash

Academic Editor

PLOS ONE

Journal Requirements:

2. We note that your paper includes detailed descriptions of individual patients/participants. As per the PLOS ONE policy (http://journals.plos.org/plosone/s/submission-guidelines#loc-human-subjects-research) on papers that include identifying, or potentially identifying, information, the individual(s) or parent(s)/guardian(s) must be informed of the terms of the PLOS open-access (CC-BY) license and provide specific permission for publication of these details under the terms of this license. Please download the Consent Form for Publication in a PLOS Journal (http://journals.plos.org/plosone/s/file?id=8ce6/plos-consent-form-english.pdf). The signed consent form should not be submitted with the manuscript, but should be securely filed in the individual's case notes. Please amend the methods section and ethics statement of the manuscript to explicitly state that the patient/participant has provided consent for publication: “The individual in this manuscript has given written informed consent (as outlined in PLOS consent form) to publish these case details.

Reviewers' comments:

Reviewer's Responses to Questions

**Comments to the Author**

1. Is the manuscript technically sound, and do the data support the conclusions?

Reviewer #1: Yes

2. Has the statistical analysis been performed appropriately and rigorously? 

Reviewer #1: Yes

3. Have the authors made all data underlying the findings in their manuscript fully available?

Reviewer #1: Yes

4. Is the manuscript presented in an intelligible fashion and written in standard English?

Reviewer #1: Yes

5. Review Comments to the Author

Reviewer #1: Title:

1. Given that there were no DR nor nephropathy, I suggest making the tone of the title simpler, for example.

" Prevalence of diabetes in pregnancy and microvascular complications in native Indonesian women: the Jogjakarta diabetic retinopathy initiatives in pregnancy (Jog-DRIP) "

Methods/Results:

1. Please confirm all the participants had OGTT. I understand that subjects with known diabetes should not have this test? Please describe how you conducted for the subjects with known diabetes.

2. Authors reported the prevalence of DR, which was none. At the same time, I was wondering how were OCT parameters such as retinal thickness in the macula. Given this study is more focused in DR, it would be informative to share the profile of OCT parameters. It might be the case that authors aiming to publish this in the following papers...?

Discussion:

1. Potential reasons for not detecting any DR in this study sample should be discussed clearly.

Authors mentioned that the cohort is younger than DiabCare's cohort. Yes, older age is a major risk factor.

Another major factor is duration of diabetes. Cases identified in this study had shorter duration of diabetes, which means mostly newly diagnosed diabetes? Proportion of known diabetes and newly diabetes in this cohort? Is this comparable to other studies?

2. How do authors consider the sample size estimation in this study? Does this needs to be updated to detect less frequent events? Discussion and perspectives for the future study would be helpful.

6. PLOS authors have the option to publish the peer review history of their article (what does this mean?). If published, this will include your full peer review and any attached files.

Reviewer #1: No

---

## [Author Response · Author response to Decision Letter 0]

29 Sep 2021

Editor Comments: The paper should be checked by a professional speaker of English before complete acceptance.

Response: One of our co-authors, A/Prof. Lim, is a native English speaker and she has reviewed this manuscript.

1. Please ensure that your manuscript meets PLOS ONE's style requirements, including those for file naming. The PLOS ONE style templates can be found at https://journals.plos.org/plosone/s/file?id=wjVg/PLOSOne_formatting_sample_main_body.p df and https://journals.plos.org/plosone/s/file?id=ba62/PLOSOne_formatting_sample_title_authors_ affiliations.pdf

Response: We have followed the style requirements.

2. We note that your paper includes detailed descriptions of individual patients/participants. As per the PLOS ONE policy (http://journals.plos.org/plosone/s/submission-guidelines#loc- human-subjects-research) on papers that include identifying, or potentially identifying, information, the individual(s) or parent(s)/guardian(s) must be informed of the terms of the PLOS open-access (CC-BY) license and provide specific permission for publication of these details under the terms of this license. Please download the Consent Form for Publication in a PLOS Journal (http://journals.plos.org/plosone/s/file?id=8ce6/plos-consent-form- english.pdf). The signed consent form should not be submitted with the manuscript, but should be securely filed in the individual's case notes. Please amend the methods section and ethics statement of the manuscript to explicitly state that the patient/participant has provided consent for publication: “The individual in this manuscript has given written informed consent (as outlined in PLOS consent form) to publish these case details.

Response: All data provided in this manuscript were anonymous. We have ensured that there was no identifiable personal information in this report. Written consent to participate in this study and subsequent publication of any data pertaining to this study was obtained from each participant and kept secured. We have revised our statement in the method section as suggested.

Page 4, line 85-87:” Written consents to participate in this study and to report all relevant data in the subsequent publications were obtained from all participants, and for those who were unable to see, read, or write, verbal consents were obtained.”

Response: We have now included our questionnaire (in both the original language and English) as Supporting Information (S1 and S2 Appendix).

Response: We have now corrected information in the ‘Funding Information’ and are requesting to update our ‘Financial Disclosure’ statement to include the grant numbers as shown below.

Financial Disclosure: “This study was supported by the Australia-Indonesia Institute in the Australian Government’s Department of Foreign Affairs and Trade in the form of funding awarded to MBS (AII2018/19073), and the Indonesian Endowment Fund for Education (LPDP) Ministry of Finance in the form of a grant awarded to FW (20161012049462). The funders had no role in study design, data collection and analysis, decision to publish, or preparation of the manuscript.”

Reviewers' comments: Reviewer #1: 

Title:

1. Given that there were no DR nor nephropathy, I suggest making the tone of the title simpler, for example.

" Prevalence of diabetes in pregnancy and microvascular complications in native Indonesian women: the Jogjakarta diabetic retinopathy initiatives in pregnancy (Jog-DRIP) "

Response: We agree and thank the reviewer for the suggestion. We have now changed the title as suggested.

Page 1, line 1-3: “Full title: Prevalence of diabetes in pregnancy and microvascular complications in native Indonesian women: the Jogjakarta diabetic retinopathy initiatives in pregnancy (Jog-DRIP)”

Methods/Results:

1. Please confirm all the participants had OGTT. I understand that subjects with known diabetes should not have this test? Please describe how you conducted for the subjects with known diabetes.

Response: All participants who never been diagnosed with diabetes underwent OGTT. However, as mentioned in page 6, line 134-136, there were 13 women who vomited during the glucose load and refused to continue. For these women, the diagnosis of diabetes was determined only based on their fasting blood glucose level.

For women who were diagnosed with diabetes prior to pregnancy, diabetes diagnosis was confirmed by their family physician following the American Diabetes Association criteria: had fasting plasma glucose 37.0 mmol/L or random glucose 311.1 mmol/L or A1C 3 6.5%. We have now added this information.

Page 6, line 124-127: “For participants who were diagnosed with diabetes before their pregnancy, diabetes diagnosis was confirmed by their family physician following the American Diabetes Association criteria [2], thus; OGTT was not performed.”

2. Authors reported the prevalence of DR, which was none. At the same time, I was wondering how were OCT parameters such as retinal thickness in the macula. Given this study is more focused in DR, it would be informative to share the profile of OCT parameters. It might be the case that authors aiming to publish this in the following papers...?

Response: We obtained OCT images for participants who were confirmed having TDP. However, none of these participants had any macular abnormalities identified from their OCT images. We have now added this in the results.

Page 9, line 209 - 210: “The median CSFT of participants with TDP was 236.5 μm (IQR 232.5 - 243).”

Discussion:

1. Potential reasons for not detecting any DR in this study sample should be discussed clearly.

Authors mentioned that the cohort is younger than DiabCare's cohort. Yes, older age is a major risk factor.

Another major factor is duration of diabetes. Cases identified in this study had shorter duration of diabetes, which means mostly newly diagnosed diabetes? Proportion of known diabetes and newly diabetes in this cohort? Is this comparable to other studies?

Response: Our study did not document any DR. We have discussed this in the discussion (please refer to page 11, line 236 - 245). The reason was mainly because our participants had shorter duration of diabetes compared to participants from other studies. We have also provided comparisons with other studies in the discussion (please refer to page 11, line 236 - 245).

To our knowledge, no study has presented data on the proportion of DIP and PDM in their study population due to its relatively new classifications. We have now made this clear in the discussion (page 11, line 224 - 225). Even in the key paper by Guariguata et al. (2014), the proportion of DIP and PDM in their reported TDP prevalence was not reported. Therefore, we tried to compare our results with previous studies that reported DR prevalence in either reported DIP or PDM group (page 11, line 236-245).

Page 11, line 224 - 225: “Moreover, there were no studies that have reported data on the proportion of DIP and PDM.”

Page 11, line 236 - 245: "We did not document any evidence of DR in this study cohort. A study by Sugiyama and associates also reported that the prevalence of DR among pregnant women in Japan was very low, where only 1.2% of pregnant women with overt diabetes (which had a similar definition with the current DIP) had DR [22]. In contrast, Rasmussen and colleagues, who studied Danish pregnant women with type 2 diabetes, showed that DR was significantly higher, at 14% of 110 pregnant women [23]. This discrepancy might be due to the shorter duration of diabetes in our study (a median of 6 months) compared to Rasmussen’s study (a mean of 3.3 years for the no progression group and 6.7 years for the progression group). Another reason might be due to the small number of TDP cases in our cohort that decreased our ability to detect DR."

2. How do authors consider the sample size estimation in this study? Does this needs to be updated to detect less frequent events? Discussion and perspectives for the future study would be helpful.

Response: Our sample size was estimated using the formula for prevalence studies with multi-stage cluster sampling (approximate number of pregnancies in Jogjakarta: 49,000 [10], using the precision of estimates to identify prevalence of hyperglycaemia in pregnancy at 27% [4]). A minimum sample size of 630 pregnant women (around 30 from each cluster) was sufficient to demonstrate the prevalence with sufficient power and precision (95% confidence interval [95% CI] 21%, 32%) (Please refer to page 5, line 103 - 108).

There were very few studies reporting the prevalence of TDP among pregnant women. Therefore, our sample size estimation was based on the previously available evidence that report the prevalence of diabetes in pregnancy, but not DR. Because the prevalence of TDP was very small in our population, our sample size could have been underpowered to detect DR in this population. Sample size estimation for future studies should consider small events of TDP and diabetes complications among TDP. We have our discussion in this regard.

Page 12 - 13, line 269 – 278: “Thirdly, due to the small number of TDP cases detected in our study population, this study could have been underpowered to measure a precise rate of DR and nephropathy in pregnant women with TDP. Sample size estimation for future studies should consider small events of TDP and diabetes complications among TDP. Finally, due to the impact of COVID-19 pandemic throughout 2020, postpartum follow-up examinations could not be completed; thus, the outcome of the diabetes status of women with DIP and DR and nephropathy status in those with TDP after delivery could not be determined. Future studies with long period of follow-up covering postpartum period and the subsequent pregnancies would capture better picture related to the course of TDP and development of diabetes complications among TDP.”

---

## [Decision Letter · Decision Letter 1]

13 Apr 2022

Prevalence of diabetes in pregnancy and microvascular complications in native Indonesian women: the Jogjakarta diabetic retinopathy initiatives in pregnancy (Jog-DRIP)

PONE-D-21-21657R1

Dear Dr. Sasongko,

We’re pleased to inform you that your manuscript has been judged scientifically suitable for publication and will be formally accepted for publication once it meets all outstanding technical requirements.

Kind regards,

Muhammad Furqan Akhtar

Academic Editor

PLOS ONE

Additional Editor Comments (optional):

Reviewers' comments:

Reviewer's Responses to Questions

**Comments to the Author**

1. If the authors have adequately addressed your comments raised in a previous round of review and you feel that this manuscript is now acceptable for publication, you may indicate that here to bypass the “Comments to the Author” section, enter your conflict of interest statement in the “Confidential to Editor” section, and submit your "Accept" recommendation.

Reviewer #1: All comments have been addressed

2. Is the manuscript technically sound, and do the data support the conclusions?

Reviewer #1: Yes

3. Has the statistical analysis been performed appropriately and rigorously? 

Reviewer #1: Yes

4. Have the authors made all data underlying the findings in their manuscript fully available?

Reviewer #1: Yes

5. Is the manuscript presented in an intelligible fashion and written in standard English?

Reviewer #1: Yes

6. Review Comments to the Author

Reviewer #1: I have no further comments on this manuscript. This study is simple, but still holding an important message reflecting the current diabetes care in microvascular complications in pregnant women.

7. PLOS authors have the option to publish the peer review history of their article (what does this mean?). If published, this will include your full peer review and any attached files.

Reviewer #1: No

---

## [Editor Report · Acceptance letter]

6 Jun 2022

PONE-D-21-21657R1 

Prevalence of diabetes in pregnancy and microvascular complications in native Indonesian women: the Jogjakarta diabetic retinopathy initiatives in pregnancy (Jog-DRIP) 

Dear Dr. Sasongko:

I'm pleased to inform you that your manuscript has been deemed suitable for publication in PLOS ONE. Congratulations! Your manuscript is now with our production department. 

Kind regards, 

on behalf of

Dr. Muhammad Furqan Akhtar 

Academic Editor

PLOS ONE